# Fatal Myocarditis following COVID-19 mRNA Immunization: A Case Report and Differential Diagnosis Review

**DOI:** 10.3390/vaccines12020194

**Published:** 2024-02-13

**Authors:** Pedro Manuel Barros de Sousa, Elon Almeida Silva, Marcos Adriano Garcia Campos, Joyce Santos Lages, Rita da Graça Carvalhal Frazão Corrêa, Gyl Eanes Barros Silva

**Affiliations:** 1University Hospital of the Federal University of Maranhão, Barão de Itapari Street 227, São Luís 65020-070, MA, Brazil; pedro.manuel.1@ebserh.gov.br (P.M.B.d.S.);; 2Clinical Hospital of Botucatu Medical School, São Paulo State University, Professor Mário Rubens Guimarães Montenegro Avenue, Botucatu 18618-687, SP, Brazil; 3Department of Pathology, Ribeirão Preto Medical School, University of São Paulo, Ribeirão Preto 14049-900, SP, Brazil

**Keywords:** COVID-19, immunization, mRNA vaccine, myocarditis, rheumatic heart disease, cathecolamin-induced cardiotoxicity, multisystem inflammatory syndrome

## Abstract

Carditis in childhood is a rare disease with several etiologies. We report a case of infant death due to pericarditis and myocarditis after the mRNA vaccine against COVID-19 (COVIDmRNAV). A 7-year-old male child received the first dose of the COVIDmRNAV and presented with monoarthritis and a fever non-responsive to oral antibiotics. The laboratory investigation showed signs of infection (leukocytosis, high levels of c-reactive protein). His condition rapidly deteriorated, and the patient died. The autopsy identified pericardial fibrin deposits, hemorrhagic areas in the myocardium, and normal valves. A diffuse intermyocardial inflammatory infiltrate composed of T CD8+ lymphocytes and histiocytes was identified. An antistreptolysin O (ASO) dosage showed high titers. The presence of arthritis, elevated ASO, and carditis fulfills the criteria for rheumatic fever. However, valve disease and Aschoff’s nodules, present in 90% of rheumatic carditis cases, were absent in this case. The temporal correlation with mRNA vaccination prompted its inclusion as one of the etiologies. In cases of myocardial damage related to COVID-19mRNAV, it appears to be related to the expression of exosomes and lipid nanoparticles, leading to a cytokine storm. The potential effects of the COVID-19mRNAV must be considered in the pathogenesis of this disease, whether as an etiology or a contributing factor to a previously initiated myocardial injury.

## 1. Introduction

Carditis in childhood is a rare disease with a variable clinical presentation, sometimes non-specific and spontaneously resolving, with the possibility of developing sequelae, which are occasionally severe and fatal [1,2,3]. Precise etiological investigation is crucial for therapeutic management as different pathogenic mechanisms guide medication choice, with endomyocardial biopsy and histological evaluation being the current gold standard. [4].

The most common causes of myocarditis are bacterial and viral infection, systemic inflammatory disorders involving connective tissue, autoimmunity, and the effects of drugs and toxins [3]. Recently, an already-known pathogen has gained even more attention: severe acute respiratory syndrome coronavirus 2 (SARS-CoV-2) [5,6]. Conditions often related to multisystem inflammation have also been reported, albeit less frequently, following coronavirus disease 2019 (COVID-19) vaccination, primarily developed using viral mRNA [7,8,9,10,11,12].

We present a case of infant death from an unknown pathology that began after the use of an mRNA vaccine against COVID-19. The autopsy identified pericarditis and myocarditis, with extensive morphological overlap between the possible differential diagnoses and a difficult final etiological classification.

## 2. Case Report

A 7-year-old male child presented with myalgia and fever 3 days after receiving the first dose of the COVID-19 vaccine (BNT162b2). The parents denied previous contact with sick people or a history of symptoms of upper airway infection. After seven days, he presented with monoarthritis in his right ankle. Blood tests indicated leukocytosis, but a CT scan of the ankle showed no abnormalities. He was diagnosed with septic arthritis and discharged with empirical antibiotic therapy.

After 10 days, the patient was re-evaluated due to persistent symptoms. At this time, leukocytosis had improved, and O antistreptolysin (OAS) values were normal, leading to discharge with a new outpatient antibiotic regimen. Three weeks post-symptom onset, with ongoing joint pain and walking difficulty, new tests were performed. Elevated OAS levels prompted hospital admission for intravenous antibiotic treatment with oxacillin. The joint pain improved over nine days, but the patient developed mild gastrointestinal symptoms like vomiting with blood streaks and epigastric pain. His condition rapidly deteriorated upon diagnosis of upper gastrointestinal bleeding. Post-orotracheal intubation, active bleeding was observed from the tube, leading to an emergency department referral. Laboratory tests indicated leukocytosis, elevated C-reactive protein levels, and negative COVID-19 polymerase chain reaction (Table 1), alongside right upper and lower lung lobe consolidations and ground-glass opacities on chest CT. Unfortunately, the patient died.

On autopsy, external examination showed anasarca and increased abdominal volume. The internal organs showed diffuse edema, including the brain, with pleural, pericardial, and peritoneal cavitary effusion. The heart exhibited a granular, opaque, whitish external surface, akin to fibrin deposits on the pericardium (Figure 1).

The myocardium had a soft consistency, alternating between pale and hemorrhagic areas, while the valves remained preserved. Microscopy revealed disseminated vascular thromboembolism. The macroscopic and microscopic findings of the main organs are reported below (Table 2, Figure 2, Figure 3 and Figure 4).

The condition was diagnosed as acute pericarditis and myocarditis, without valve involvement, with heart failure leading to pulmonary edema, complicated by acute tubular necrosis and ischemic hepatic necrosis.

## 3. Discussion

This case poses a diagnostic challenge due to overlapping risk factors, symptoms, and diverse histological findings, each with variable diagnostic specificity. The main differential diagnoses will be discussed.

### 3.1. Rheumatic Fever (RF) and Myocarditis

The previous occurrence of an upper airway infection associated with arthritis and elevated OSA levels would fulfill the modified Jones criteria for diagnosing RF, with the presence of one major and two minor criteria: carditis, fever, and elevated serum CRP levels, respectively. However, evidence of monoarthritis is not a diagnostic criteria, as only polyarthritis is considered in diagnosing a first outbreak [13].

Furthermore, the morphological presentation differs from that typically seen in cardiac involvement by rheumatic fever. Most rheumatic carditis involves the endocardium [14], with valve disease in up to 90% of symptomatic cases [13]. Associated pericarditis and myocarditis, when present, show morphological characteristics reflecting the pathogenic mechanisms involved.

Cross-immune activation through antigenic mimicry with streptococcal proteins leads to the systemic inflammatory damage characteristic of RF [14,15]. Other studies have shown integration between streptococcal proteins and type IV collagen in the extracellular matrix [16]. In the heart, this reaction is more accentuated at the endothelium, especially in the valves, with increased expression of VCAM-1, an adhesion molecule that helps in the migration of activated leukocytes [17]. Moreover, the inflammatory aggregates are arranged around the cardiac connective tissue, intermingling the muscle fibers, without pronounced myocardial necrosis [18], which, when present, is related to local cellular aggression caused by the inflammatory process. This presentation differs from virus-related carditis, where aggression primarily affecting the myocardium results in extensive necrosis and a corresponding rise in myocardial necrosis markers [3].

The most specific histopathological finding of rheumatic heart disease (RHD) is Aschoff’s nodules, perivascular histiocyte aggregates with characteristic nuclear changes [14]. Spina et al. identified the frequency of this finding in endomyocardial biopsies ranging from 19 to 67% (average: 41.8%). However, the studies failed to establish a consistent relationship with prognosis, corticoid use, or β-hemolytic Streptococcus prophylaxis [19].

### 3.2. Viral Carditis

Enteroviruses are classically associated with viral myocarditis. Over time, new entities have gained importance, such as parvovirus B19, influenza, adenovirus, cytomegalovirus, human immunodeficiency virus, and SARS-CoV-2 [4], accompanied by different pathogenic mechanisms. Adenoviruses and enteroviruses possess a cytolytic action profile that damages the myocardial cytoskeleton and is possibly linked to CCR5 receptor expression [20]. Parvovirus B19 exhibits vasculotropism and can remain quiescent in endothelial cells, causing damage to myocytes through inflammatory stimuli [4].

The expanding evidence on SARS-CoV-2 cardiotoxicity reveals various pathogenic mechanisms, including cardiomyotropism and cell damage via the angiotensin-converting enzyme receptor-binding protein, immune activation by the spike protein, and the production of antibodies that cross-react with cardiac cell antigens like α-myosin [21]. After a collaborative systematic review, Almamlouk found that 100% of studies show an association between cardiac infection with SARS-CoV-2 and myocardial necrosis, while there is no reference to signs of myocarditis, such as a pronounced inflammatory infiltrate [22]. The study failed to define a histological lesion pattern associated with COVID-19. Notably, a systematic review identified cardiomegaly, myocardial necrosis, an inflammatory infiltrate composed of CD3^+^ T lymphocytes, with prominent CD8^+^, and macrophages as the main cardiac signs identified. [5]. The absence of a clear relationship between viral load and cell damage, myocardial necrosis, and the low frequency of organized and pronounced inflammatory infiltrates make it less likely that the mechanism involved in COVID-19 is cytotoxic injury. The vasculitis caused by the virus, including arterial damage and occlusion, along with the systemic effects of the infection, such as adrenergic response and cathecolamin-induced cell stress, may be key contributors to its harmful effects [23,24,25,26].

### 3.3. Vaccination against COVID-19 and Myocarditis

The general population’s use of vaccines, following their safety confirmation in phase 3 studies, increases exposure and enables the identification of rarer side effects. This was also true for the COVID-19 vaccine, especially the viral mRNA-based one [27,28].

Vaccine-related myocarditis is one of these adverse effects. The Adverse Event Reporting System (VAERS) included 27,229 cases of myocarditis and pericarditis until June 2023 [29]. With an often favorable clinical course, several studies corroborate the higher frequency of this complication after the second dose in young males under 40 years of age [30], especially in the 18–25 age group, with a higher risk attributed to the mRNA-1273 vaccine than to the BNT162b2 [31]. However, studies show that the booster dose does not lead to a substantial increase in the risk of perimyocarditis [21].

Giannotta et al. described the mechanisms involved in cardiac injury stimulated by the mRNA vaccine. The stimulation of the expression of exosomes, containing both the spike viral protein and inflammatory mediators, associated with the expression of adhesion factors that dysfunctionally stimulate the endothelial cell, plays an important role in this mechanism [7,32]. The spike protein leads to activation of the TLR-4/NF-kB pathway and stimulation of the cell-mediated immune response, with inflammation directed at cardiomyocytes [33]. In addition to the effect related to the viral structure, the composition and quantity of lipid nanoparticles in the vaccine dose, which differ between manufacturers, can show toxic activity with a potent inflammatory response already in the first moments after application [34]. There is also evidence that the immune cells that absorb the lipid nanoparticles distribute them throughout the body with high levels of spike protein, inflicting a continuous immune response [29]. The immune reaction comprises CD8^+^ T lymphocytes, macrophages, and plasma cells, occasionally including an eosinophilic component without a characteristic morphological pattern [35].

Post-vaccination inflammatory activation is evidenced by a storm of inflammatory cytokines, such as high levels of IL-1, IL-1B, IL-6, and TNF-α. The circulation of these mediators might relate to the development of side effects and individual reactions after the first vaccine dose, but more frequently after the second dose, with varying clinical significance [35,36].

### 3.4. Multisystem Inflammatory Syndrome (MIS)

Multisystem inflammatory syndrome (MIS) is a condition related to COVID-19, with a predilection for children (MIS-C) [37]. The diagnostic criteria defined by the World Health Organization [38] include fever > 3 days, increased markers of inflammation, no evidence of other infections, and proof of COVID-19 infection, in addition to two of the following criteria: rash, non-purulent conjunctivitis, or mucocutaneous inflammation; hypotension or shock; myocardial dysfunction, pericarditis, or valvulitis; coagulopathy; and gastrointestinal symptoms. Diaz et al. identified a series of 35 children with defined criteria for a diagnosis of MIS-C, all with cardiac involvement. In another series of eight children with hyperinflammatory syndrome and probable COVID-19 infection, seven had gastrointestinal symptoms on initial presentation, as well as fever for 4 to 5 days [39].

Although rare, cases of MIS have also been reported after vaccination against COVID-19 (MIS-V) without evidence of concomitant virus infection [40]. Wassif et al. reported 10 cases of perimyocarditis related to COVID-19 vaccination, including 1 case associated with MIS, marked by a significant reduction in left ventricular function and requiring intensive treatment [41]. Ourdali identified 12 cases of MIS among over 4 million vaccinated children aged 12 to 17 with mRNA vaccines, with cardiac involvement in 83% of cases. Gastrointestinal symptoms (83%) and cytolytic hepatitis (50%) were also common [9].

### 3.5. Diagnostic Considerations

This is a case with complex clinical laboratory findings. Myocarditis and pericarditis, only suspected at the time of autopsy, developed in an indolent and nonspecific manner, which led to difficulty in raising this hypothesis for its appropriate investigation. This highlights the importance of investigating deaths with undefined causes. For comparison, Table 3 summarizes the histological findings of the diagnostic hypothesis.

The patient met the criteria defined for RHD. Nevertheless, some clinical and morphologic details raised suspicion for another causative or contributing factor since they differed from the classic presentation of rheumatic fever. The patient did not report a clinical history of streptococcal infection, despite the fact that Jones modified criteria acknowledge the possibility of subclinical infection if there is laboratorial evidence (i.e., high OSA levels). In addition, there were also criteria for MIS-C in the case, since myocardial disfunction, coagulopathy, and gastrointestinal symptoms developed related to fever > 3 days and an increase in inflammatory markers.

Previous exposure to streptococcal strains with immunogenic potential is an important risk factor for myocarditis. Still, the lack of typical findings in rheumatic heart disease, such as Aschoff nodules, even with extensive histological examination, makes the clinical pathological correlation difficult. A series of endomyocardial biopsies found a considerable prevalence of this finding, regardless of the limited material. In addition, the absence of valve disease is uncommon in RHD, reaching 10% of cases [19]. Unfortunately, the complementary methods for the detection of viral mRNA in the cardiac tissue were not feasible at the time of investigation.

## 4. Conclusions

It is reasonable to address the potential effects of the COVIDmRNAv in this setting. Temporal relationships must be evaluated carefully since they do not evoke a causal relationship. However, the growing evidence of the vaccine’s systemic immunological effects allows us to deduce the possibility of the contribution of the cytokine storm to the establishment of myocardial injury, already initiated by rheumatologic mechanisms. The systemic findings developed by the patient are similar to those of MIS, which can be frequently present in patients with post-COVID-19 vaccine myocarditis.

## Figures and Tables

**Figure 1 vaccines-12-00194-f001:**
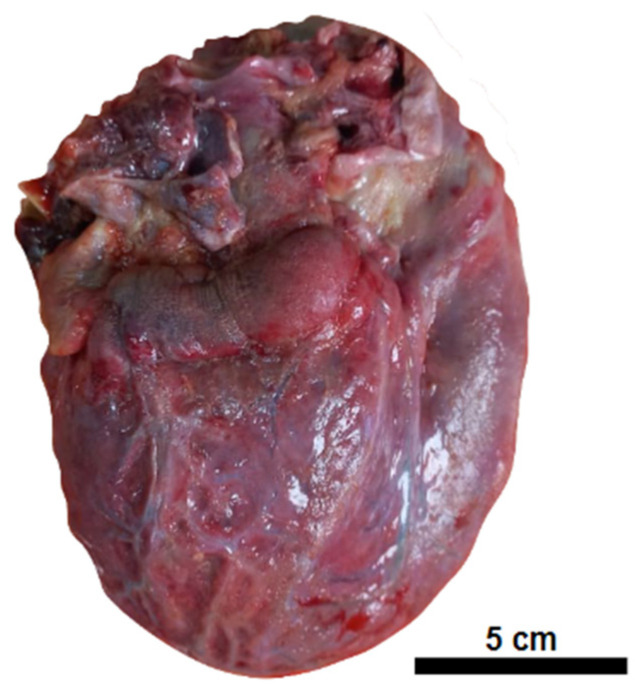
Pericarditis. Globose heart with an opaque external surface covered in fine granulation and fibrinous debris.

**Figure 2 vaccines-12-00194-f002:**
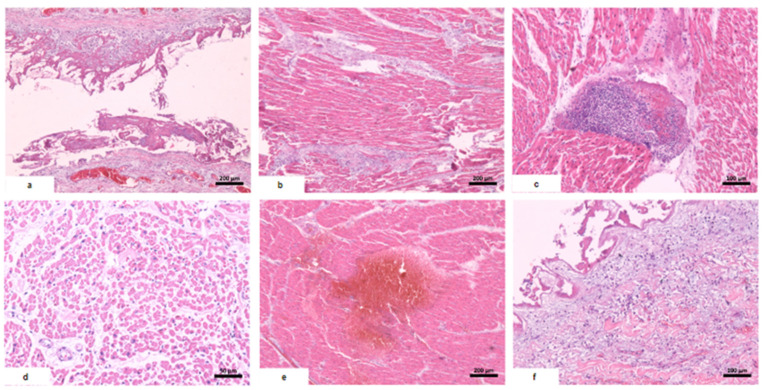
Histological alterations in the heart. (**a**) Pericarditis: fibrin deposition in the pericardium and lymphocytic infiltrate. (**b**) Myocarditis: inflammatory infiltrate concentrated in intermyocardial fibrotic tracts, with focal extension to cardiac fibers. (**c**) Foci of mixed inflammatory aggregate in the myocardium: plasma cells, lymphocytes, and neutrophils. (**d**) Subendocardial necrosis: myocardial fibers with eosinophilic and vacuolized cytoplasm and absent nuclei. (**e**) Myocardial hemorrhage. (**f**) Focal endocarditis: discrete mixed inflammatory infiltrate in the endocardium with fibrin deposition.

**Figure 3 vaccines-12-00194-f003:**
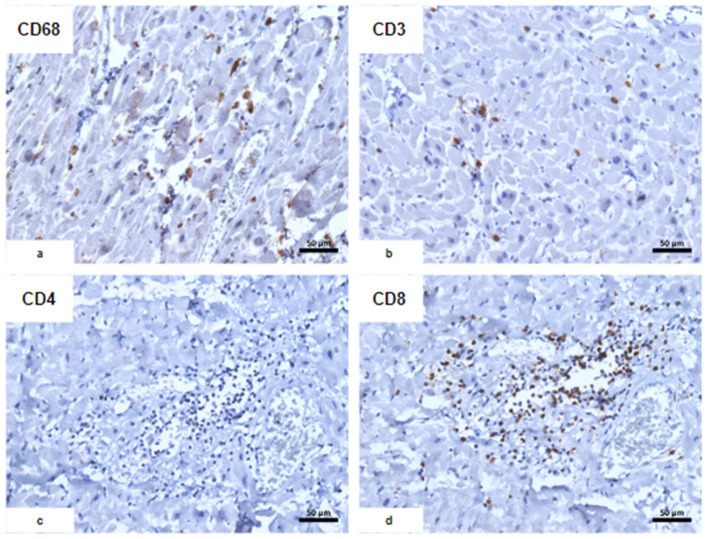
Immunohistochemical characterization of the inflammatory infiltrate. (**a**) CD68; (**b**) CD3; (**c**) CD4; (**d**) CD8: predominance of CD8+ T lymphocytes associated with CD68+ macrophages with an usual morphological appearance.

**Figure 4 vaccines-12-00194-f004:**
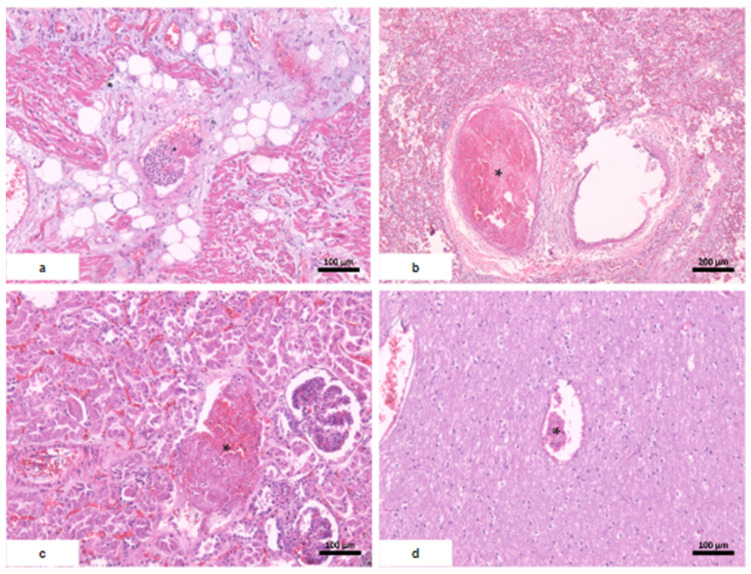
Systemic thromboembolism. (**a**) Heart; (**b**) lungs; (**c**) kidney with severe acute tubular necrosis; (**d**) brain. * fibrin thrombi.

**Table 1 vaccines-12-00194-t001:** Laboratory tests during the course of the disease.

	Day 7	Day 17	Day 21	Day 31
Hemoglobin (g/dL)	-	9.24	9.5	9.87
Hematocrit (%)	-	28.1	26.7	28.3
Leukocytes (cells/μL)	170,000	14,600	15,000	28,900
Segmented (%)	76.0	-	-	-
Neutrophils (%)	76.9	59.2	59.6	64.1
Lymphocytes (%)	-	29	32.9	2.6
Platelets (mm^3^)	-	453,000	467,000	445,000
OSA (U/mL) *	-	4.1	1425.2	
C-reactive protein (mg/dL)	-	-	8.38	13.21
PT/INR **	-	-	10.9/1.00	
aTTP ***	-	-	29.9	
Urea (mg/dL)	-	-	24.69	31.72
Creatinine (mg/dL)	-	-	0.74	0.41
Sodiu (mmol/L)	-	-	138	-
Potassium (mol/L)	-	-	4.7	-
CPK (U/L) ****	-	-	-	49.97

* O antistreptolysin; ** prothrombin time/international normalized ratio; *** activated partial thromboplastin time; **** creatine phosphokinase; not available.

**Table 2 vaccines-12-00194-t002:** Pathological findings of the autopsy procedure.

	Brain	Heart	Lungs	Liver	Rins
Weight (g)	1362	298	Right: 404 Left: 304	1368	Right: 116 Left: 134
Macroscopic findings	Edema and tonsil herniation.	Fibrinous perimyocarditis with hemorrhage. Valves unchanged	Edema and congestion.	Moderate steatosis hemorrhagic appearance	Signs of shock.
Microscopic findings	Tissue edema. Reactive gliosis. Vascular thromboembolism.	Pericardium: intense deposition of fibrin, red blood cells, and inflammatory cells. Myocardium: interfascicular edema, coagulative myocyte necrosis. Hemorrhage and inflammatory infiltrate rich in macrophages and t-cells. Aschoff’s nodules and/or Anitschkow’s cells: not observed. Endocardium: slight loose fibrosis and inflammatory infiltrate.	Parenchymal hemorrhage. Vascular thromboembolism. Inflammatory infiltrate of lymphocytes, neutrophils, and xanthomatous macrophages.	Hemorrhagic necrosis. Vascular thromboembolism. Microvesicular steatosis.	Acute tubular necrosis.

**Table 3 vaccines-12-00194-t003:** Differential diagnosis of carditis.

	Rheumatic Fever	COVID-19	Multisystem Inflammatory Syndrome (MIS)	Post COVID-19 mRNAV
Diagnostic criteria	Revised Jones criteria *	Presence of cardiac involvement (pericarditis, myocarditis, or valvulitis) during proven COVID-19 infection.	World Health Organization **	Temporal relationship + exclusion of other causes.
Presentation	Valvulitis in 90% of cases. Myocardium and pericardium are occasionally affected.	Cardiomegaly, Variable clinic, cardiac dysfunction when associated with MIS.	Fever. Presence of multisystem involvement (gastrointestinal, central nervous system). Carditis in >80% of cases.	Favorable evolution when not associated with MIS. Predilection for young men.
Mechanism	Cross-inflammatory reaction centered on connective tissue.	Spike protein, endothelial activation, macrophage activation.	Lymphocytes and macrophages, neutrophils, and eosinophils possible.	Lymphocytes and macrophages, neutrophils, and eosinophils possible
Inflammatory infiltrate	Inflammatory infiltrate in scattered collagenous bands, with lymphocytes and macrophages.	Variable. Can be discrete and unorganized or in a pattern of myocarditis.	Not described	Lymphocytes and macrophages, neutrophils, and eosinophils possible.
Myocardial necrosis	Discrete, related to local inflammatory infiltrate.	Frequent.	--	Present, focal.
Specific findings	Aschoff’s nodules in up to 67% of cases.	Absent	Absent.	Absent.
Vascular effects	Aortic involvement in 20% of cases of mitral valve disease.	Vasculitis and thromboembolism in the microvasculature.	Not reported.	--

* two major criteria, or one major and two minor criteria. Major: carditis, polyarthritis, chorea, erythema marginatum, subcutaneous nodule. Minor: polyarthralgia, fever, elevated ESR and/or CRP, increased PR; ** fever > 3 days, increased markers of inflammation, no evidence of other infections, proof of COVID-19 infection, and two of the following criteria: rash, non-purulent conjunctivitis or mucocutaneous inflammation; hypotension or shock; myocardial dysfunction, pericarditis, or valvulitis; coagulopathy; gastrointestinal symptoms.

## Data Availability

No new data were created or analyzed in this study. Data sharing is not applicable to this article.

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
