# Peer review of "Fatal Myocarditis following COVID-19 mRNA Immunization: A Case Report and Differential Diagnosis Review"

_vaccines, 2024, doi:10.3390/vaccines12020194_

Round 1

Reviewer 1 Report

Comments and Suggestions for Authors

1.This case report provides information about fatal myocarditis in children caused by COVID-19 vaccine(COVIDmRNAV), which is of great value for timely diagnosis and treatment of serious complications induced by COVID-19 vaccine(COVIDmRNAV).

2. Since the report did not provide exact evidence to prove the mechanism of myocarditis caused by COVID-19 vaccination, it is inappropriate to propose a possible mechanism in the abstract: Myocardial damage related to COVID-19mRNAV appears to be related with the expression of exosomes and lipid nanoparticles leading to a cytokine store.

The focus of this case report should be on why the author diagnosed this case as myocarditis caused by COVID-19 vaccine(COVIDmRNAV).

3.There is no specific data for Monocytes (%) in Table 1, why is it still listed?

4.Please standardize the punctuation in the caption of Figure 3 and Figure 4.“Figure 3:(a). CD68 (b). CD3 (c). CD4 (d). CD8:”“Figure 4: (a). Heart. (b). Lungs. (c). Kidney with severe acute tubular necrosis. (d). Brain. ”

5.Please use a three line chart for all tables.

6.The reference numbers after the 6th article appear twice.

7.The conclusion should be concise, and relevant content can be integrated into the discussion section.

Comments on the Quality of English Language

Minor editing of English language required.

Author Response

We appreciate your acknowledgment on the relevance of this report.

In this sentence from the abstract, we provide the current understanding on myocardial effects of the COVIDmRNAV, not specifically related to the case reported. This phrase will be reconstructed in order to make it more clear that we are not affirming that this is the cause of the case, but the findings in literature so far.

The tables design, the line in table 1, the punctuation in figures 3 and 4 captions and the reference numbers will be corrected, it was missed in our revision. The conclusion will be rearrenged in order to fit its objective in the report. We also included text related to a new reference that was found.

The additions made are highlited in red and the section rearrenged is underlined.

Sincerely,

Pedro Manuel Barros de Sousa

Reviewer 2 Report

Comments and Suggestions for Authors

Dear Authors!

COVID-19 infection  is a severe life-treatening disease especially in adults with comorbidities. Vaccination prevent the disease severity and decrease the number of fatal outcomes in world.

In children the disease is milder. The main severe complication is a multisystem inflammatory syndrome.

Vaccination of children can prevent the overall incidence and decrease the transmission of virus from kids to adults

If vaccination against COVID works against the risk of MIS-C is still unclear.

The cases of MIS-C associated with SARSCoV2 vaccination was described. 

Information about the safety of the vaccine is very improtant.

The main bias part is related that vaccine-associated etiology is not fully confirmed.

I have several suggestions:

Table 1 required the full name of all abbreviations in the footnotes. What does it mean OSA?

Please provide the reference intervals for tests in the 1st column.

Add the ferritin and D-dimer levels to the table 1.

Do you possibility to find viral RNA (vaccine antigens) in the tissues to confirm the etiology?

Did you check viruses in the blood or in the tissues, especially in heart with tissue PCR or with immunohistochemistry assay

Was the patient corresponded to the criteria of MIS-C, which set of criteria? 

The discussion is good, structurated

Please add the limitation section at the end of the discussion section

Author Response

Dear reviewer!

We are very pleased to know you see the importance in this case report.

The corrections solicited in Table 01 will be made. Unfortinatelly, the ferritin and d-dimer levels were not informed/assessed by the clinical team.

The COVID-19 test that was negative is the PCR of nasal swab sample.

The child met the WHO criteria for MIS-C, if we remove the COVID-19 evidence of infection. They are listed on the table 2 caption, but we will add a paragraph to the discussion making it more evident.

We stablished contact with different research groups in order to perform additional testing for viral RNA detection on the cardiac tissue. Unfortunately, none of then performed such technics. We tried performing immunohistochemistry in the formalin fixed parafin embeded material, with poor results.

The conclusion will be rearrenged in order to fit its objective in the report, as it was request by another reviewer. We also included text related to a new reference that was found.

The additions made are highlited in red and the section rearrenged is underlined.

Sincerely,

Pedro Manuel Barros de Sousa

Round 2

Reviewer 2 Report

Comments and Suggestions for Authors

Dear Authors! Thank you for your comments.

I have no additional queries